# Multidisciplinary Postoperative Validation of ^18^F-FDG PET/CT Scan in Nodal Staging of Resected Non-Small Cell Lung Cancer

**DOI:** 10.3390/jcm11237215

**Published:** 2022-12-05

**Authors:** Benedetta Bedetti, Philipp Schnorr, Sarah May, Jürgen Ruhlmann, Hojjat Ahmadzadehfar, Markus Essler, Alexander Quaas, Reinhard Büttner, Joachim Schmidt, Holger Palmedo, Yon-Dschun Ko, Kai Wilhelm

**Affiliations:** 1Department of Thoracic Surgery, Helios Clinic Bonn/Rhein-Sieg, 53123 Bonn, Germany; 2Department of Radiology and Nuclear Medicine, Johanniter Hospital Bonn, 53113 Bonn, Germany; 3Department of Nuclear Medicine and Radiology, Medicine Centre Bonn, 53115 Bonn, Germany; 4Department of Nuclear Medicine, Klinikum Westphalen, 44309 Dortmund, Germany; 5Institute of Pathology, University Hospital Cologne, 50937 Köln, Germany; 6Division of Thoracic Surgery, Department of General, Thoracic and Vascular Surgery, Bonn University Hospital, 53127 Bonn, Germany; 7Institute of Radiology and Nuclear Medicine, Kaiser-Passage Bonn, 53113 Bonn, Germany; 8Department of Oncology, Johanniter Hospital Bonn, 53113 Bonn, Germany

**Keywords:** PET/CT scan, nodal staging, lymph node metastases, non-small cell lung cancer

## Abstract

Background: The aim of this study was to examine the validity of PET/CT scans in the preoperative identification of lymph node metastases (LNM) and compare them with postoperative outcomes. Methods: In this retrospective study, we included 87 patients with a solitary lung nodule or biopsy-proven non-small cell lung cancer treated in our institution from 2009 to 2015. Patients were divided into two groups and four subgroups, depending on pre- and postoperative findings. Results: According to our analysis, PET/CT scan has a sensitivity of 50%, a specificity of 88.89%, a positive predictive value of 63.16%, and a negative predictive value of 82.35%. Among the patients, 13.8% were downstaged in PET-CT, while 8% were upstaged. In 78.2% of cases, the PET/CT evaluation was consistent with the histology. Metastases without extracapsular invasion were seldom recognized on PET/CT. Conclusions: This analysis showed the significance of extracapsular tumor invasion, which causes an inflammatory reaction, on LNM, which is probably responsible for preoperative false-positive findings. In conclusion, PET/CT scans are very effective in identifying patients without tumors. Furthermore, it is highly probable that patients with negative findings are free of disease.

## 1. Introduction

Lung cancer is one of the most common causes of death worldwide. Together with tumor size and local extension, the presence of lymph node metastases (LNM) is a pivotal prognostic factor that influences the therapeutic path and its success rate [1]. Staging and therapy for patients with non-small cell lung cancer (NSCLC) are currently based on the Eighth Edition of the Lung Cancer Stage Classification [2]. ^18^F-fluorodeoxyglucose positron emission tomography/computed tomography (^18^F-FDG PET/CT) is a fundamental diagnostic tool that is routinely used preoperatively to perform clinical staging. In particular, PET/CT scans play a cardinal role in the diagnosis of lymph node status [3]. In a meta-analysis, Hellwig at al. showed that the diagnostic accuracy of ^18^F-FDG PET/CT scan in detecting mediastinal LNM in lung cancer patients had a discrepancy of 12 ± 2% in sensitivity and 8 ± 1% in specificity compared with the final histological findings [4]. The aim of this study was to address this problem and thus demonstrate the validity of ^18^F-FDG PET/CT in preoperatively identifying LNM compared to postoperative outcomes.

## 2. Materials and Methods

All procedures performed in this retrospective observational study involving human participants were in accordance with the ethical standards of our institutional research committee and the 1964 Declaration of Helsinki and its later amendments. Formal institutional consent is not required for this type of study.

Our institution is a certified lung cancer center, and it is standard within the preoperative preparation to obtain patient consent to use data for research purposes.

### 2.1. Patients

In this retrospective study, we included 87 patients with a solitary lung nodule or biopsy-proven non-metastatic NSCLC that were surgically treated at the Lung Cancer Centre Bonn/Rhein Sieg from 2009 to 2015. All patients received a preoperative staging diagnosis according to current guidelines, including ^18^F-FDG PET/CT. Every clinical case was discussed by our interdisciplinary tumor board, and the treatment plan was defined according to presurgical tumor stage. Patients who had undergone induction chemotherapy and/or radiation were excluded. Data were provided after patient consent was obtained.

### 2.2. PET/CT Scan

All PET/CT scans were performed using a Siemens Biograph PET/CT system in three cooperating nuclear medicine units (Siemens Biograph 2 PET/CT; Siemens Biograph Sensation 16 PET/CT; Siemens Biograph 64 True Point).

^18^F-2-fluoro-2-deoxy-D-glucose (^18^F-FDG) was used as a tracer in all patients. Blood glucose levels were measured before performing the scan, and the examination was conducted only in patients with a glucose level <150 mL/dL.

After injecting 300–382 MBq ^18^F-FDG during the distribution phase, the patients were placed in a resting position to avoid any muscular activity. The PET/CT scan was performed 60–90 min after tracer injection. The CT scan was performed as a low-dose multi-slice CT with 2, 16, or 64 slices. The entire examination was conducted according to current guidelines.

### 2.3. Interdisciplinary Tumor Board (MDT)

All patients were discussed during our weekly interdisciplinary tumor board. PET/CT scans were performed when the solid tumor component measured more than 8 mm [5]. All patients with PET-positive lymph nodes underwent preoperative EBUS or invasive mediastinal staging.

### 2.4. Mediastinal Lymphadenectomy—Surgical Technique

All patients underwent surgery at the Department of Thoracic Surgery, Helios Clinic Bonn/Rhein-Sieg. Every patient underwent anatomical tumor resection together with systematic mediastinal en bloc lymph node resection, either open or thoracoscopically. Lymph nodes in stations 2R, 4R, 7, 8, and 9 were harvested for right-sided cancers and stations 4L, 5, 6, 7, 8, and 9 for left-sided cancers. The specimens were then fixed in formalin, labeled using a standard histology sheet (Table 1), and sent to pathology for macro- and microscopic analysis.

### 2.5. Histopathological Examination

Description of the lymph node specimens was made according to Mountain and Dresler [6]. Histopathological examinations were performed by the Institute of Pathology, Cologne University Hospital.

Macroscopic evaluation described the number and size of the harvested lymph nodes. All specimens were then dehydrated and embedded in paraffin. They were cut into 4 µm thin sections and placed on microscope slides. The paraffin-embedded sections were stained with hematoxylin–eosin according to standard practice.

### 2.6. Data Analysis and Postoperative Validation

Pre- and postoperative tumor stages were compared after receiving the histopathological lymph node findings. Patients were divided into two groups, with each group separated into two subgroups, as follows (Table 2):
Group 1 included patients with pathologically confirmed lymph node involvement. Subgroup 1.1 showed no discrepancy between presumed pre- and confirmed postoperative lymph node involvement, whereas subgroup 1.2 patients were PET-negative but had pathologically positive lymph nodes (false negative, Figure 1a,b).Group 2 consisted of patients with no pathological detection of infiltrated lymph nodes. Subgroup 2.1 showed no discrepancy, but subgroup 2.2. included patients with suspected increased glucose uptake on PET/CT (indicating preoperatively presumed lymph node involvement) but without histopathological confirmation (false positive, Figure 1b,c).

We calculated the sensitivity, specificity, and positive and negative predictive values for the detection of LNM through PET/CT.

Pre- and postoperative tumor stages were evaluated in groups 1.2 and 2.2 to determine up- and downstaging.

All of the lymph nodes of these two groups were microscopically re-evaluated to uncover possible reasons for false-negative/false-positive findings on PET/CT based on the following criteria:Maximum size and number of lymph nodes per station;Distribution of metastasis in a single lymph node (Figure 2)-Focal metastasis-Subtotal nodular metastasis (>50% of the extent of the lymph node)-Multifocal nodular metastases

Diffuse metastases;

3.Tumor spread (within the lymph node or extracapsular).

Upon further examination, it was hypothesized that small LNM without extracapsular invasion were difficult to detect on PET/CT scan, whereas diffuse metastases had a higher chance of being observed.

The lymph node samples from group 2.2 were microscopically re-evaluated to determine the possible causes of false-positive findings based on PET/CT; they were especially inspected following tissue reactions, including sarcoid-like lesions, silico-anthracosis, and lymphofollicular hyperplasia. All lymph node specimens were sliced into thinner layers, stained with hematoxylin-eosin (HE), and immunohistochemically processed with anti-cytokeratin AE1/AE3 antibodies. In this way, we could rule out the possibility that some lymph node metastases remained undetected in the previous examination. At the same time, the PET/CT scans of the patients in this group were reviewed together with the radiologists to check whether there was a correlation between the maximum standardized uptake value (SUVmax) of the primary tumor and the presence of LNM. It was postulated that patients with LNM should have higher SUVmax values than patients without LNM.

### 2.7. Statistical Analysis

Statistical analysis of the data was performed in cooperation with the Bonn University Statistical Institute. Fisher’s exact test was used to assess the correlation between groups. Two-dimensional contingency tables were used to calculate the sensitivity, specificity, and predictive values, referring to the PET/CT scans. Multidimensional frequency tables were used for testing hypotheses. Graphic presentation was executed through scatter charts. Data analysis and collection were performed using Excel 2021 (Microsoft, Redmond, WA, USA)

## 3. Results

Between 2009 and 2015, 310 patients (136 females, 174 males) with NSCLC were diagnosed and treated at the Lung Cancer Centre Bonn/Rhein Sieg. Ninety-six patients underwent surgery without induction chemotherapy, but only 87 patients were included in the final analysis (Figure 3), as the PET/CT findings or histological specimens were not available for nine patients. The patient groups are shown in Table 2.

Histology

The different histology types were distributed similarly among the four groups (Table 3).

### 3.1. Results in Subgroup 1.2

Size

Group 1.2 included 12 patients with negative PET/CT finding but with LNM in the final histopathology report (false-negative findings). Overall, 31 LNM were discovered (Table 4).

The detection threshold on the PET/CT scans was 4 mm [7]. Of the lymph nodes studied, 41.94% were smaller than 4 mm, and 58.06% were equal to or larger than 4 mm. A significant correlation was found between the LNM diameter being equal to or larger than 4 mm and detection on the PET/CT scan (*p* < 0.02). Furthermore, these two variables were in linear correlation with one another (Figure 4).

Localization

LNM localization was analyzed, and the results are shown in Table 5. The hypothesis that diffuse LNM could be detected at a higher rate on PET/CT scans could not be proven.

Extracapsular LNM

Table 6 shows the frequency of extracapsular LNM in the analyzed lymph nodes. Metastases without extracapsular invasion were seldom recognized on PET/CT; this was statistically significant (*p* < 0.05).

### 3.2. Results in Subgroup 2.2

Subgroup 2.2 included seven patients with positive PET/CT findings but without LNM in the final histopathology report (false-positive findings). In total, 15 LNM that were detected on PET/CT were not confirmed by histopathology. The secondary histopathological review of these lymph node stations led to the identification of different tissue reactions, as reported in Table 7. These findings were compared with patients in group 2.1, in which 55 lymph node stations were reviewed overall.

Silico-anthracosis was present in 9% of the stations and lymphofollicular hyperplasia was present in 31% of the stations. According to our evaluation, it could not be proven that lymph nodes with lymphofollicular hyperplasia showed PET positivity more often than lymph nodes without any tissue reaction (*p* = 0.2).

### 3.3. Impact of SUVmax on the Detection of LNM on PET/CT

The SUVmax related to the primary tumor on PET/CT scan was measured for every patient. In group 1 patients (*n* = 24), the mean average SUVmax was 11.14 (range: 1.7–21.1), and in group 2 patients (*n* = 63), it was 10.42 (range: 1.8–30.9). The difference between the mean SUVmax values of the two groups was not statistically significant (*p* = 0.39).

### 3.4. Comparison between Pre- and Postoperative TNM Classification

Table 8 and Table 9 show the TNM classification of patients in groups 1.2 and 2.2. Downstaging of LNM based on PET/CT findings was observed in 13.8% of patients, and upstaging was observed in 8% of patients. Therefore, preoperative PET/CT and histopathological findings matched in 78.2% of the patients.

### 3.5. Detection of LNM by ^18^F-FDG PET/CT

Table 10 shows the assessment of LNM by PET/CT of all 87 patients. According to our analysis, ^18^F-FDG PET/CT scans have a sensitivity of 50%, a specificity of 88.89%, a positive predictive value (PPV) of 63.16%, and a negative predictive value (NPV) of 82.35%.

Overall, 13.8% of the patients were downstaged in PET/CT, and 8% were upstaged. In 78.2% of the cases, the PET/CT evaluation matched the histological evaluation.

## 4. Discussion

Lung cancer treatment is a major health challenge worldwide, as the diagnosis is often made at an advanced phase due to a lack of symptoms in the early stages, so therapeutic possibilities are limited by that point [8]. Therefore, the implementation of an early and accurate staging process is essential for delivering the best therapeutic options to every patient. Together with clinical, laboratory, pathological, and molecular genetic investigations, diagnostic imaging plays an essential role in patient identification and staging. Conventional and nuclear radiology methods allow clinicians to characterize the tumor’s extent; in particular, PET/CT scans can reveal the functional metabolic activity of the tumor and its possible metastases, both local and distant [9]. Since they are obviously larger, PET-positive lymph nodes can have a benign origin, while small lymph nodes can contain micrometastases, which are rarely detected preoperatively [10,11]. This background helped us establish the aim of our study, as we attempted to define the accuracy of ^18^F-FDG PET/CT in identifying LNM. We analyzed possible causes of false-positive and false-negative findings in nodal status based on pathological and nuclear medicine criteria. Numerous publications have stated that lesions larger than 4 mm can easily be detected through ^18^F-FDG PET/CT scan [7,12]. In our study, 41.94% of the LNM, which were not identified through PET/CT, were smaller than 4 mm; therefore, metastasis size may explain these mismatched results. On the other hand, 58.06% of the non-identified LNM were larger than 4 mm. A possible explanation for the high false-negative rate, despite the presence of lymph nodes larger than 4 mm, is the small size of the primary tumor and thus lower associated SUVmax [13,14,15,16]. In addition, high blood glucose levels and the proximity of the LNM to the primary tumor are factors that can lead to false-negative results. Other possible reasons for a false-negative evaluation of the nodal status are the tracer’s brief distribution time and the SUV threshold value being too high [12].

Our analysis showed that 6.7% of the PET-positive LNM in group 1.2 had a diameter smaller than 4 mm. This probably correlates with a simultaneous inflammatory reaction taking place in those lymph node stations, which showed higher tracer uptake as a result [13]. In fact, in two patients in subgroup 1.2, all LNM were smaller than 4 mm, which in theory should not have displayed PET positivity based on size alone [6]. In the other patients in subgroup 1.2, at least one LNM was larger than 4 mm and should have been effectively detected on the PET scan.

Another postulated hypothesis is that certain LNM configurations are less likely to be detected by PET scans; for example, focal metastases may be less likely to show PET positivity than diffuse LNM. However, the correlation between these factors could not be statistically confirmed, probably because of the small sample size (*n* = 12).

In our analysis, LNM without extracapsular invasion was significantly less likely to be detected on PET/CT. Wenzel et al. published a study in 2004 regarding the significance of extracapsular tumor invasion on LNM of tumors of the upper digestive tract and upper airways [17]. They reported that lymph node tumor invasion caused an immunological reaction, which led to fibrotic alteration of the lymph nodes. This may also be applicable to lung cancer LNM. It is conceivable that an immunological process in the lymph nodes could cause a magnified metabolic reaction, which would lead to increased uptake on PET/CT. However, we should point out that our analysis is based on a small sample size (12 patients and a total of 31 LNM); these results would probably be even more evident in a large-scale study.

Upon analyzing the results of subgroup 2.2, we found sarcoid-like lesions in 6.7% of the lymph nodes. Sarcoid-like lesions can develop in tumors and in tumor-draining lymph nodes. During an immunologic reaction, macrophages are transformed into epithelioid cells, which are involved in the formation of granulomas [18,19]. This inflammatory reaction can lead to PET positivity in the involved lymph nodes, which causes false-positive findings [12].

The same principle applies to silico-anthracosis, which was found in 33.3% of the false-positive lymph node stations. Silico-anthracosis is a granulomatous disease that can trigger a strong inflammatory reaction, which results in increased tracer uptake on PET/CT.

In many false-positive lymph nodes, we found follicular hyperplasia, which is normally caused by a nonspecific reaction of the lymph nodes with the development of secondary activated lymph follicles, in part with expansion of the T-cell zone [20]. A study by Chung et al. described an increased incidence of lymphofollicular hyperplasia in false-positive lymph nodes after PET [21]. However, we could not confirm this hypothesis in our study, as lymphofollicular hyperplasia was found in 33% of patients who had negative PET findings and no trace of LNM in the final pathology report.

A review of all pathology specimens of the false-positive patients (group 2.2) was performed to exclude the possibility that preoperative PET positivity was caused by undetected LNM.

The literature states that high uptake by the primary tumor is associated with a higher probability of metastatic spread [22,23], but this statement could not be verified by our study. In fact, we calculated the SUVmax of the primary tumor for all our patients, and we could not find any significant correlation between SUVmax and LNM.

Subedi et al. published a study in 2009 in which they investigated the preoperative staging accuracy of PET/CT scans in comparison with CT scans in 161 lung cancer patients [23]. As in our analysis, their standard reference was postoperative histopathology. The results showed that the PET/CT scans matched the histopathology in 78% of cases. Ten percent of patients had downstaging of their lymph node status, while 12% had upstaging. These results correspond with the outcome of our study, as we included a similar sample of patients. Furthermore, our results regarding specificity and positive and negative predictive values are comparable with Subedi et al.’s study, thus confirming our conclusions. However, when we compare the PET sensitivity we had in our analysis with other publications, our results are considerably below the average. For example, Subedi et al. found a sensitivity of 74.3% [24], and Shen et al. in a meta-analysis from 2017 found a sensitivity of 65% [25]. Other authors have reported even higher values [5,26]. The probable reason for our outcome is the high number of false-negative results in our data. In fact, LNM smaller than 4 mm, which could not be detected on PET/CT scan, were particularly noticeable in these patients.

Regarding PPVs, many studies describe values as high as 89% and 95% [7,27]. In contrast, the results of Cerfolio’s large 2004 study [26] found a PPV of 46%. Due to the great discrepancy in PPVs, further studies are needed to validate these findings.

Recent studies have stated that the introduction of quantitative criteria for N staging might improve stratification of patients and thereby improve treatment and survival rates [28,29], so this aspect should be further investigated in future studies.

This study has certain limitations. First, since we performed a retrospective data analysis, causal relations cannot be deduced, so we can only consolidate or develop new hypotheses. Second, patients who had undergone induction therapy or were not candidates for surgery were excluded from the study, which could have introduced selection bias. Finally, we had a small number of patients, so the subgroups were small, which may have limited the statistical significance of some results. Therefore, further studies with more patients are warranted.

## 5. Conclusions

The aim of our study was to validate the application of ^18^F-FDG PET/CT scans in the nodal staging of patients with resectable NSCLC in the preoperative setting. Our analysis identified the significance of extracapsular tumor invasion in LNM. A consequent inflammatory reaction could cause tracer uptake in the affected lymph nodes on PET/CT, which may explain the preoperative false-positive findings. Another significant result was the identification of LNM smaller than 4 mm in almost half of the patients with false-negative findings, which could explain their preoperative PET negativity. In conclusion, ^18^F-FDG PET/CT scans are very significant in identifying patients without tumors (specificity). Furthermore, patients with negative findings have a high probability of indeed being free from disease (NPV). Finally, our study confirms the fundamental role of PET/CT in the preoperative diagnosis of LNM in patients with NSCLC, particularly in identifying patients without LNM.

## Figures and Tables

**Figure 1 jcm-11-07215-f001:**
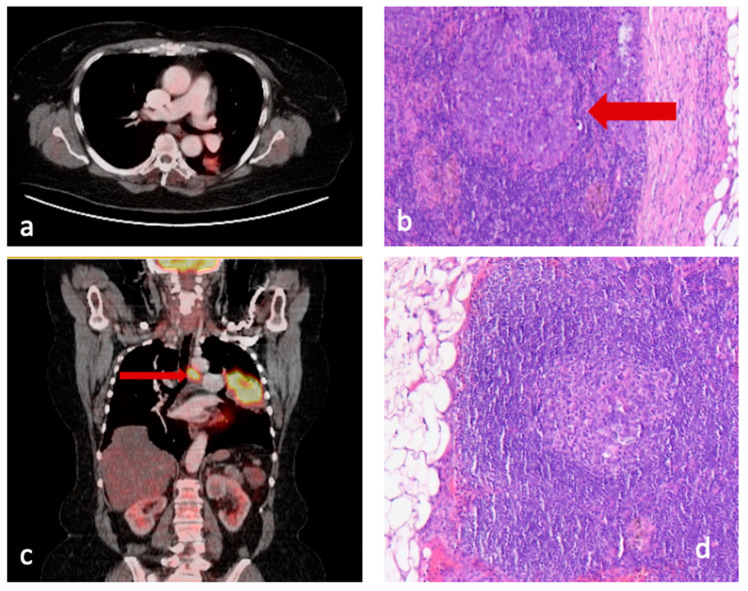
(**a**) PET/CT scan of patient in group 1.2, where there is no suspected LNM; (**b**) Pathological specimen of same patient with HE stain showing metastasis; (**c**) PET/CT scan of patient in group 2.2 with suspected LNM; (**d**) Pathological specimen of same patient with HE stain showing a lymph node free of disease.

**Figure 2 jcm-11-07215-f002:**
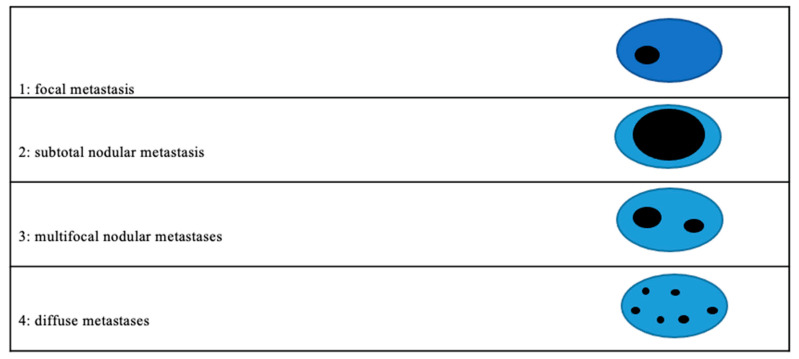
Pattern of metastases distribution in the lymph nodes.

**Figure 3 jcm-11-07215-f003:**
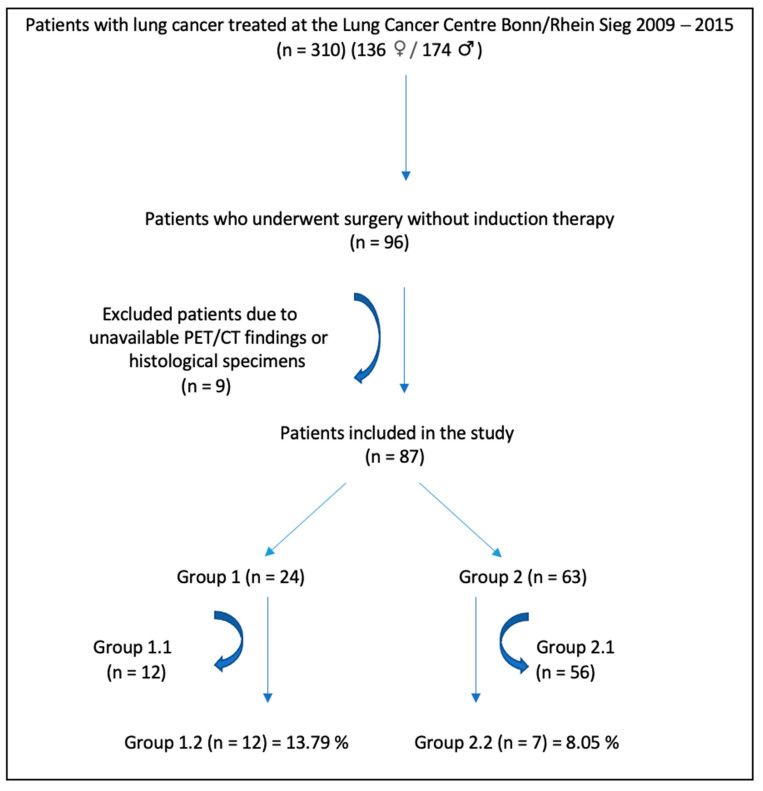
Flow chart for data evaluation.

**Figure 4 jcm-11-07215-f004:**
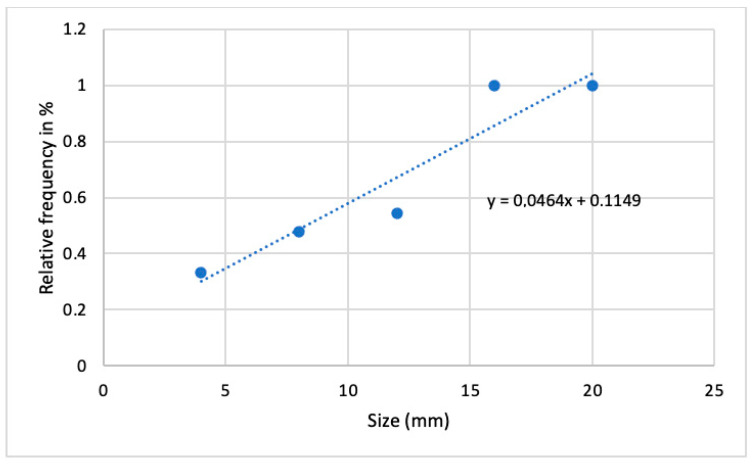
Linear correlation between metastasis size and detection frequency on PET/CT scan.

**Table 1 jcm-11-07215-t001:** Standard histology sheet for localization of mediastinal lymph nodes.

Station	N2	Description	Station	N	Description
1	N2	Highest mediastinal	8	N2	Paraesophageal
2	N2	Upper paratracheal	9	N2	Pulmonary ligament
3	N2	Prevascular and retrotracheal	10	N1	Hilar
4	N2	Lower paratracheal (including azygos nodes)	11	N1	Interlobar
5	N2	Subaortic (aortopulmonary window)	12	N1	Lobar
6	N2	Para-aortic (ascending aorta or phrenic)	13	N1	Segmental
7	N2	Subcarinal	14	N1	Subsegmental

**Table 2 jcm-11-07215-t002:** Patient classification.

Group 1 (24 patients) Pathologically confirmed lymph node involvement	Group 1.1 (12 patients) ^18^F-FDG PET/CT positive and histopathology positive → No discrepancy between presumed pre- and confirmed postoperative lymph node involvement Group 1.2 (12 patients) ^18^F-FDG PET/CT negative and histopathology positive → discrepancy (False negative)
Group 2 (63 patients) Lymph nodes free from disease after pathological examination	Group 2.1 (56 patients) ^18^F-FDG PET/CT negative and histopathology negative → no discrepancy Group 2.2 (7 patients) ^18^F-FDG PET/CT positive and histopathology negative → discrepancy (False positive)

**Table 3 jcm-11-07215-t003:** Histology in the four groups.

Group (Number of Patients)	Adenocarcinoma	Squamous Cell Carcinoma
1.1 (12)	6	6
1.2 (12)	7	5
2.1 (56)	29	27
2.2 (7)	3	4

**Table 4 jcm-11-07215-t004:** Number and size of LNM in group 1.2.

Patients in Group 1.2	Number LNM (n)	LNM Size (mm)
1	1	1
2	1	4
3	1	10
4	2	4; 2
5	6	2.5; 3; 2 × 4; 2 × 5
6	1	5
7	10	1.5; 2; 3 × 3; 5; 2 × 6; 8; 11
8	1	3
9	1	7
10	2	1; 1.5
11	1	11
12	4	1.5; 4; 5; 8

**Table 5 jcm-11-07215-t005:** Localization of LNM not detected on PET/CT in group 1.2.

Pattern of LNM Distribution (Figure 2)	Number (n)	Rate (%)
1	7	22.58
2	12	38.71
3A	0	0
3B	3	9.68
4	9	29.03

**Table 6 jcm-11-07215-t006:** Extracapsular invasion in group 1.2.

Extracapsular Invasion	Number (*n*)	Rate (%)
Present	9	29.03
Absent	22	70.97

**Table 7 jcm-11-07215-t007:** Lymph node tissue reactions in comparison between groups 2.2 and 2.1.

Lymph Node Tissue Reaction	Number of Stations Involved (*n*)		Rate (%)	
	Group 2.2	Group 2.1	Group 2.2	Group 2.1
Sarcoid-like lesions	1	0	6.7	0
Silico-anthracosis	5	5	33.3	9
Lymphofollicular hyperplasia	7	17	46.7	31
Normal tissue	2	33	13.3	60

**Table 8 jcm-11-07215-t008:** TNM classification in group 1.2.

cTNM (Preoperative)	pTNM (Postoperative)
cT1b/cN0/cM0 → IA	pT1b/pN1/cM0 → IIA
cT2b/cN0/cM0 → IIA	pT3/pN1/cM0 → IIIA
cT1a/cN0/cM0 → IA	pT1b/pN1/cM0 → IIA
cT2a/cN0/cM0 → IB	pT2a/pN1/cM0 → IIA
cT2a/cN0/cM0 → IB	pT3/pN2/cM0 → IIIA
cT2b/cN0/cM0 → IIA	pT2b/pN1/cM0 → IIB
pT2a/cN0/cM0 → IB	pT2b/pN2/cM0 → IIIA
pT2b/cN0/cM1b → IV	pT2a/pN1/cM0 → IIA
cT1b/cN0/cM1b → IV	pT1b/pN1/cM0 → IIA
cT1a/cN0/cM0 → IA	pT2a/pN2/cM0 → IIIA
cT2b/cN0/cM0 → IIA	pT2b/pN1/cM0 → IIB
pT1b/cN0/cM0 → IA	pT1b/pN1/cM0 → IIA

**Table 9 jcm-11-07215-t009:** TNM classification in group 2.2.

cTNM (Preoperative)	pTNM (Postoperative)
cT2b/cN1/cM0 → IIB	pT2b/pN0/cM0 → IIA
cT2b/cN2/cM1a → IV	pT2b/pN0/cM1a → IV
cT2a/cN2/cM0 → IIIA	pT2b/pN0/cM0 → IIA
cT2b/cN1/cM0 → IIB	pT1a/pN0/cM0 → IA
cT2a/cN1/cM0 → IIA	pT2a/pN0/cM0 → IB
cT1b/cN1/cM0 → IIA	pT1a/pN0/cM0 → IA
cT3/cN1/cM0 → IIIA	pTx/pN0/cM0 → nicht bestimmbar

**Table 10 jcm-11-07215-t010:** Detection of LNM by PET/CT.

^18^F-FDG PET/CT in Correlation with Pathology Findings	Number of Patients
Positive (^18^F-FDG PET/CT positive and histopathology positive)	12
False positive (^18^F-FDG PET/CT positive and histopathology negative)	7
Negative (^18^F-FDG PET/CT negative and histopathology negative)	56
False negative (^18^F-FDG PET/CT negative and histopathology positive)	12

## Data Availability

The data underlying this article cannot be shared publicly to protect the privacy of the individuals who participated in the study. The data will be shared on reasonable request to the corresponding author.

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
