# Peer review of "Multidisciplinary Postoperative Validation of 18F-FDG PET/CT Scan in Nodal Staging of Resected Non-Small Cell Lung Cancer"

_jcm, 2022, doi:10.3390/jcm11237215_

Round 1

Reviewer 1 Report

SUMMARY

The current retrospective observational study from a single institute’s experience is aimed to define the accuracy of 18F-FDG PET/CT in the identification of lymph node metastasis (LNM) preoperatively and valid of PET/CT scans using the standard reference the postoperative histopathology. The authors reported that the preoperative PET/CT and histopathological findings matched in 78,2 % of the patients;  13,8% of patients, who were downstaged in PET-CT, and 8%, who were upstaged. Accordingly, the study had a sensitivity of 50 %, a specificity of 88.9 %, a PPV of 63.2 %, and an NPV of 82.4%. They concluded that a significant relevance of the extracapsular tumor invasion in the LNM and LNM smaller than 4 mm in almost half of the patients with false negative findings.

As the authors mentioned, obviously larger, PET-positive lymph nodes can also have a benign origin and small lymph nodes can contain micrometastases were well-known in the lung cancer fields. Several comments below might be strengthened the manuscript:

1.      Please provide ethical approval from your hospital.

2.      Please explain the conditions in which how to include 87 patients in your study, such as consecutive or selective.

3.      Lymph node dissection technique and examination are important in the validation study to convince the true negative. Please explain these two portions in more detail in your study.

4.      Regarding LNM size and number, the majority of criticism in the literature was the lymph nodes were harvested usually piece by piece instead of the whole lymph node harvested. Please describe in your study how to avoid it.

5.      In the majority of clinical practice, patients in groups 1.1, and 2.2 who had preoperative positive lymphadenopathy from the preoperative PET/CT scan should follow by tissue proof, such as EBUS or mediastinoscopy. Please explain the approach and treatment policy in your multidisciplinary lung cancer meeting.

6.      As the authors’ mentioned in their study, selection bias was the major limitation of this study. 

Reviewer 2 Report

Dear Authors/Editors.

I read with interest the manuscript entitled “Multidisciplinary postoperative validation of 18F-FDG-PET/CT-scan in nodal staging for resected non-small cell lung cancer” in which authors evaluated in a retrospective series the effectiveness of PET/CT to detect nodal metastases in NSCLC.

Auhors divided the whole cohort in 4 group (real positive, false positive, real negative and false negative) and found a sensitivity of 50% together with a specificity of 88,89%.

However, the manuscript presents some issues and concerns that should be solved.

1)      There are several typos and grammar errors; the text should be revised by a mother tongue

2)      Introduction provides few information on the role of the PET/CT in NSCLC staging behind the nodal status

3)      In M&M details on the specimen histology should be reported; different tumors as lepidic or mucinous adenocarcinoma are usually PET/CT negative, as well as big squamous carcinoma could present a central infected necrosis that may affect the FDG uptake

4)      Uptake differences between T, N and eventual M should be reported as well as its impact on the final pStage.

5)      Have the author evaluated the difference between N1 and N2 and the role of PET/CT in the nodal skip metastases

6)      Tables should be organized differently to improve the readability of the text

7)      Authors should define the impact of their study on the routinely pre-operative assessment

8)      How many patients underwent EBUS or mediastinal invasive staging in the light of the PET/CT findings

  • Authors are invited to add in the discussion the following articles: PMID: 34039110, DOI: 10.1177/03008916211018515 and PMID: 29788107,DOI: 10.1093/icvts/ivy171

Round 2

Reviewer 2 Report

Dear authors and editors

 I really appreciated the effort made to improve the quality of the manuscript.

I have no more concerns or issue to be highlithed.